# Innate Lymphoid Cells in the Malignant Melanoma Microenvironment

**DOI:** 10.3390/cancers12113177

**Published:** 2020-10-29

**Authors:** Aintzane Apraiz, Aitor Benedicto, Joana Marquez, Andrea Agüera-Lorente, Aintzane Asumendi, Elvira Olaso, Beatriz Arteta

**Affiliations:** Department of Cell Biology and Histology, School of Medicine and Nursing, University of the Basque Country (UPV/EHU), 48940 Leioa, Spain; aintzane.apraiz@ehu.es (A.A.); aitor.benedicto@ehu.eus (A.B.); joana.marquez@ehu.eus (J.M.); andrea.aguera@ehu.eus (A.A.-L.); aintzane.asumendi@ehu.eus (A.A.); elvira.olaso@ehu.eus (E.O.)

**Keywords:** melanoma, tumor microenvironment, innate lymphoid cells, extracellular vesicles

## Abstract

**Simple Summary:**

Innate lymphoid cells (ILCs) are the innate counterparts of adaptive immune cells. Emerging data indicate that they are also key players in the progression of multiple tumors. In this review we briefly describe ILCs’ functions in the skin, lungs and liver. Next, we analyze the role of ILCs in primary cutaneous melanoma and in its most frequent and deadly metastases, those in liver and lung. We focus on their dual anti– and pro-tumoral functions, depending on the cross-interactions among them and with the surrounding stromal cells that form the tumor microenvironment (TME) in each organ. Next, we detail the role of extracellular vesicles secreted to the TME by ILCs and melanoma on both cell populations. We conclude that the identification of markers and tools to allow the modulation of individual ILC subsets, in addition to the development of standardized protocols, is essential for addressing the therapeutic modulation of ILCs.

**Abstract:**

The role of innate lymphoid cells (ILCs) in cancer progression has been uncovered in recent years. ILCs are classified as Type 1, Type 2, and Type 3 ILCs, which are characterized by the transcription factors necessary for their development and the cytokines and chemokines they produce. ILCs are a highly heterogeneous cell population, showing both anti– and protumoral properties and capable of adapting their phenotypes and functions depending on the signals they receive from their surrounding environment. ILCs are considered the innate counterparts of the adaptive immune cells during physiological and pathological processes, including cancer, and as such, ILC subsets reflect different types of T cells. In cancer, each ILC subset plays a crucial role, not only in innate immunity but also as regulators of the tumor microenvironment. ILCs’ interplay with other immune and stromal cells in the metastatic microenvironment further dictates and influences this dichotomy, further strengthening the seed-and-soil theory and supporting the formation of more suitable and organ-specific metastatic environments. Here, we review the present knowledge on the different ILC subsets, focusing on their interplay with components of the tumor environment during the development of primary melanoma as well as on metastatic progression to organs, such as the liver or lung.

## 1. Introduction

The incidence of melanoma has risen worldwide over the past decade, with approximately 132,000 new diagnoses each year, according to the World Health Organization [1]. Although it represents only 1% of all cutaneous malignancies, melanoma is the most deadly of all skin cancers [2]. Although the development of combined treatments for melanoma has decreased deaths among patients by around 65%, patients with metastatic disease still unfortunately encounter death as an irremediable fate, with a survival rate lower than 25% [3]. Targeted therapies have improved this scenario [4,5], but tumor resistance in metastatic melanoma is still of great concern [6]. In addition to this resistance, inherent in the tumor cells themselves, some initially responsive patients might develop de novo resistance driven by the complex reciprocal interactions between the tumor and its microenvironment [7], which leads to the failure of routine therapies. The mechanisms for the development of resistance against current therapies include changes in the immune-cell subsets towards immunosuppressive phenotypes and programmed cell death-ligand 1 (PD-L1) expression in the tumor [8]. Indeed, therapies that control inhibitory pathways, such as cytotoxic T-lymphocyte antigen-4 (CTLA-4) and/or programmed cell death protein-1 (PD-1) receptors, result in better outcomes for patients with advanced melanoma [4,5]. In addition to the previous ones, recently intervening with a second line of checkpoint blockade targeting PD-1, T-cell immunoglobulin and mucin domain-3 (TIM-3)and/or killer-cell immunoglobulin-like receptors (KIR) it’s being considered as a more effective strategy [9].

The tumor microenvironment (TME) may be considered an ecosystem where multiple cell types coexist. This specialized environment is composed of tumor cells, non-tumor cells, and a non-cellular compartment. The non-tumor cells include endothelial cells, fibroblasts, and innate and adaptive immune cells. The non-cellular compartment of the TME is composed of extracellular matrix (ECM) and extracellular vesicles (EVs). The components of the TME strongly interact with one another, which significantly impacts their function and the TME’s composition [10,11,12]. In this review, we focus on the innate lymphoid compartment during melanoma progression, including its interaction with tumor cells and other components of the TME.

## 2. Innate Lymphoid Cells

Over the last decade, innate lymphoid cells (ILCs) have emerged as new players in the immune TME, involved in melanoma progression and the acquisition of resistance. ILCs modulate the functions of immune cells, such as dendritic cells [13] and T cells [14]; moreover, they interact with other elements of the TME such as the ECM [15,16], endothelial cells [17,18], and fibroblasts [16]. All of these TME components are critical participants in melanoma progression and colonization of the liver [18,19] and lungs [20], among other sites.

ILCs and the cells of the adaptive immune system derive from a common progenitor in the bone marrow [19,20]. ILCs differ from other components of the adaptive immune system by their lack of rearranged antigen receptors [21,22]. Spits et al. [23] classified ILCs into three groups according to the differential expression of transcription factors and specific cytokine that they produce (Figure 1). Type 1 ILCs include helper ILC1 (ILC1s) and cytotoxic natural killer (NK) cells, mirroring Th1 cells in the adaptive immune system. Both subsets are characterized by the need for the transcription factor T-bet to develop and by the capacity for producing interferon gamma (IFNγ) and tumor necrosis factor alpha (TNFα) (Figure 1). NK cells were the first component of the innate lymphoid compartment to be discovered [24,25]. They produce cytotoxic molecules and have thus been considered central players in immune surveillance against tumors since their discovery. ILC1s resemble NK cells at the phenotypic level, but lack their cytotoxic potential and do not require the transcription factor Eomes for their development and function [26,27]. ILC1s also express considerable amounts of interleukin-7 receptor (IL-7R), while tissue-resident NK cells do not [28].

Type 2 ILCs (ILC2s) express the transcription factor GATA binding protein 3 (GATA3) and produce IL-5, IL-13, and IL-22 to restore the normal immune state, stimulating tissue repair through stromal reaction [29,30]. Type 3 ILCs express the transcription factor retinoic acid receptor-related orphan receptor gamma-t (RORγt) and produce IL-17 and IL-22 to create an acute immunologic response [31]. They can be further divided into those designated as ILC3s, which include natural cytotoxicity receptor-positive NCR(+) ILC3s (NKp46, NKp44, and NKp30) and NCR(−) ILC3s [32], and lymphoid tissue inducer (LTi) cells.

Even though these three ILC groups are well defined, they show a high degree of plasticity, being able to transdifferentiate into other ILC subsets when exposed to tissue-specific stimuli [33,34]. For example, NCR(+) and NCR(−)ILC3s, together with ILC2s, have the potential to turn into IFNγ-producing ILC1-like cells in the presence of IL-12 and IL-15, and IL-1, respectively [35,36].

The data beautifully summarized by Bal et al., (2020) shed some light on the pathways that mediate the dynamic differentiation patterns of ILCs [37]. Briefly, NK cells become ILC1s upon activation with tumor growth factor beta (TGFβ). Both ILC1s and ILC2s transdifferentiate into ILC3s upon IL-1β and IL-23 stimulation, while the complementary stimulation of ILC2s with TGFβ leads to their transdifferentiation into CD117- ILC3s [37]. Therefore, the plasticity of these immune populations complicates the picture of ILCs in both healthy conditions and pathogenesis. Based on the multiple phenotypes they may acquire, some authors have postulated the existence of more than three subsets [38], adding two other subsets—intraepithelial ILC1s and ILC progenitors (ILCPs)—to the abovementioned types of ILCs.

## 3. ILCs in the Skin, Liver, and Lung

The determination of melanoma-related alterations of ILCs requires a preliminary description of their status under tumor-free conditions. It is also worth briefly describing innate cells in the hepatic and lung compartments, both of which are homing and colonization sites for metastatic melanoma [39,40].

### 3.1. Skin

ILCs are predominantly tissue-resident cells that do not circulate in the body [41]. The skin homing of ILCs rely on surface receptors such as C-C motif chemokine receptor 10 (CCR10) and cutaneous lymphocyte-associated antigen (CLA) [42,43,44,45]. They also express additional homing-related receptors, such as C-C motif chemokine receptor 8 (CCR8) in dermal NK cells (Figure 2). All three types of ILCs are found in the skin, but the relative enrichment of the different phenotypes varies depending on the status of the organ [41,43,46] (Table 1). A detail description of ILCs in healthy skin could be found elsewhere [43,47].

NK cells produce perforin and granzyme B, which endow these cells with robust cytotoxic activity [44]. ILC2s and ILC3s might contribute to wound healing following activation by IL-33 derived from fibroblasts and endothelial and epithelial cells [44,45]. ILC3 NCR(−) subsets compose the major ILC subtype in healthy skin, while NCR(+)s predominate in psoriatic skin [47,48]. They produce IL-22 and express the surface marker CD56 [49] and express the chemokine receptor CCR10, increasing immune cell recruitment (Figure 2) [50].

### 3.2. Liver

As discussed by Liu and Zhang, a healthy liver contains abundant resident ILCs [51]. Type 1 ILCs form the dominant subset [51,52] (Table 1).

Hepatic Type 1 ILCs are involved in immune regulation with opposing roles, either favoring or inhibiting an effective local immune response [53,54]. Liver ILC1s, previously identified as liver-resident NK cells, reside in the organ in a state of equilibrium [55] and, unlike conventional NK cells, express TNF-related apoptosis-inducing ligand (TRAIL) [56,57]. Liver NKG2A^+^ CD49a^+^ ILC1s act in concert with conventional CD49b^+^ NK cells to maintain the local immunotolerance existing in the liver [57]. Type 2 and 3 ILCs represent about 5% of all the ILCs in a healthy liver (Figure 3). ILCs deregulation may lead to a wide range of liver disorders [54]. The activation and expansion of Type 2 and 3 ILCs in chronic hepatocellular stress or ischemia/reperfusion injury [51] may counteract injury and increase the levels of IL-22, a survival factor for hepatocytes [58].

### 3.3. Lungs

ILCs significantly populate the lung (Figure 4) [38,59,60], where they play a central role in organ homeostasis and repair [61,62]. Despite the presence of the three main subsets of ILCs, there is limited information regarding the ILC composition in healthy and damage human lung (Table 1). For example, CD69-CD56dimCD16^+^ NK cells account for the vast majority of NK cells in human lungs undergoing lobectomy [63], while the non-toxic ILC1 subset is characterized as CD56^−^IL-12Rβ2^+^ [61]. In addition, ILC2s and NCR(−) ILC3s predominate in healthy human lungs, while damage from chronic obstructive pulmonary disease leads to an increase in NCR(−) ILC3s and a concomitant decrease in ILC2 and NCR(+) ILC3 cells [61]. Notably, discrepancies are observed between human and mouse lungs, as ILC2s have been described as the main subset of ILCs in healthy mouse lungs [64]. Both ILC1s and ILC2s have been linked to host protection upon viral infection [65], in response to which epithelial or immune-cell-derived cytokines produce IL-4, IL-5, IL-13, and IL-9, along with amphiregulin [62,66], contributing to immune surveillance and the stimulation of lung-epithelium responses [67] ILC3s mainly act as a regulator of lung inflammation [68], with a central role in the IL-17/IL-22 axis, key to lung homeostasis [69,70].

Thus, even though the primary function of ILCs is to favor tissue repair, the mechanisms by which each subset in the lung and liver carries it out seem to be slightly different and to involve different modulators.

## 4. The Function of ILCs in Melanoma

### 4.1. ILCs in Primary Cutaneous Melanoma

The roles of ILCs in melanoma are consistent with their plasticity and their opposing functions determined by the external stimuli they receive from the TME (Table 2). Among all the ILC subsets observed in the melanoma microenvironment, those expressing the activating receptor NKp46, such as Type 1 ILCs and NCR(+)ILC3s, exert antitumor effects. NKp46 allows ILCs to recognize molecules expressed in tumor cells (e.g., ligands). The integrity and correct surface expression of NKp46 is essential for the ILC-mediated capacity to control melanoma development in mouse models (Figure 5) [71].

NK cells interaction with melanoma cells has been reviewed elsewhere [72]. Briefly, NK-cell-mediated tumor-cell cytotoxicity relies on the balance among ligands for activating receptors (e.g., NKG2D, DNAM-1, and NCRs) and those for inhibitory receptors (e.g., NKG2A and KIRs). Melanoma cells very often display some ligands for NKG2D (MICA/B and ULBP), while the presence of less-characterized NCR ligands seems to be high in early disease stages [72]. The relevance of NK cells in this field is also supported by their capability of killing melanoma cells with cancer-stem-cell-like characteristics, which are known to facilitate disease recurrence and metastatic spread [73,74]. However, the antitumor activity of NK cells may be altered promoting tumor evasion. One of the best-characterized mechanisms is the selection of melanoma cells with low expression of ligands that activate NK receptors or increased expression of KIR (inhibitory receptor)-ligand, class I MHC [72]. The implications of less-described roles of NK cells in the adaptive immune response should also be taken into account. NK cells, as well as pre-mNK cells (murine pre-mature NK)—functionally comparable to human CD56^bright^ HLA-DR^+^ NK cells [75]—present class II MHC at the cell surface under certain conditions [76,77], such as in contact with tumor cells.

Interestingly, Terme et al., described opposite roles for pre-mNK cells in terms of their capacity to prime CD4+ and CD8+ T cells, favoring T-cell maturation [77]. In fact, Wilson et al., observed enhanced CD4+-mediated tumor (melanoma) rejection upon the depletion of the B220^+^NK1.1 pre-mNK cell population [78]. As suggested by the authors, the negative effect of pre-mNK cells on CD4+-mediated antitumor activity could be related to T-cell exhaustion or transformation into regulatory T cells upon exposure to pre-mNK-presenting class II MHC.

In line with these results, two independent studies highlight the inverse correlation between the enrichment of the circulating CD56^bright^ NK-cell population and the survival of melanoma patients [79,80]. Moreover, activated CD11c^+^NK1.1^+^NHCII^high^ IKDC (murine pre-NK) cells were able to express the inhibitory PD-L1 and produce IL-10 [81], suppressing anti-melanoma immunity [78]. Indeed, NK cell cytotoxicity against tumor cells increases after PD-1 blockage in NK cells [82]. The mechanisms regulating the immune response to therapy could be mediated by a modulation of ILC2 function but not ILC1 nor ILC3 [74]. Since in the context of autoimmune diseases, a large percentage of CD4^+^ T cells may be killed via TNF-related apoptosis-inducing ligand (TRAIL) or perforin and granzyme B, leading to CD4^+^ T-cell exhaustion [83,84], it is tempting to hypothesize that these mechanisms operate to promote tumor development.

Little is known about the specific role of ILC1s in the melanoma context. Ercolano et al. have described an increased presence of ILC1s in melanoma patients but a functional impairment of IFNγ and TNFα [85] in the melanoma microenvironment. On the other hand, chronic tumor-cell exposure to IFNγ—secreted by Type 1 ILCs, among others—has been proposed as a relevant regulator of immune checkpoints. IFNγ induces the expression of PD-L1 on the tumor-cell surface and therefore blocks NK/ILC1-cell maturation and activity [86]. In this scenario, often detected in melanoma, Benci et al., showed that the blockage of tumor IFNγ signaling favors IFNγ production by CD8 T-cell populations. Ultimately, antitumor effects take place via either surface-antigen/MHC-I recognition or the promotion of Type-1-ILC maturation. Given the opposite roles that IFNγ may play, it would be of interest to analyze the biological consequences of the findings of Ercolano et al. [85].

The limited literature on the contribution of ILC2 cells to melanoma development points towards a dual role for this population. As shown by Long et al. [87], IL33-responsive intratumoral ILCs correlate well with functional ILC2s, Ref. [87]. The interaction of IL-33-stimulated ILC2s with NK cells resulted in an impaired ability of NK cells to eliminate B16 melanoma cells. According to these results, the depletion of ILC2s within the tumor led to an increase in NK-cell infiltration and tumor rejection (Figure 5) [87]. As stated by Wagner et al., the capacity of ILC2s to suppress tumor growth upon IL-33 stimulation was diminished by tumor-cell-derived lactic acid [88]. In vitro, the number of growing B16F10 melanoma cells inversely correlated with proliferation, survival, and IL-5 production by ILC2s. However, Wagner et al. did not decipher the underlying mechanism. Therefore, it cannot be excluded the participation of tumor C-X-C chemokine receptor type 2 (CXCR2). IL-33-activated ILC2s sustain CXCR2 expression on tumor cells by producing high amounts of CXCR2 ligands, which, in turn, induce apoptosis in malignant cells [89]. However, ILC2s also up-regulate the expression of PD-1 in CXCR2-expressing NK cells which might interfere with the antitumoral effect of ILC2 during melanoma progression.

Three facts suggest a protumoral function of ILC3s in melanoma. First, they are active producers of IL-17 [70,90,91]. IL-17 spurs B16-F10 melanoma growth by promoting angiogenesis and the recruitment of immunosuppressive populations, such as myeloid-derived suppressor cells (MDSCs) [92]. Moreover, it contributes to tumor development through IL-6 and Stat3 activation. Thus, targeting IL-17 might represent a potentially effective therapeutic approach in melanoma [93]. Second, large numbers of ILC3s exist in skin diseases, with significant inflammatory activity [94]. Third, the enrichment of the ILC3 population has been linked to tumor progression or metastasis [95,96].

#### ILCs and the Stromal Compartment in the Primary Melanoma

As part of the immune cells, ILCs participate in the creation of a complex and dynamic TME [97]. The healthy dermis is composed of large amounts of ECM, consisting mainly of collagen, elastin, and proteoglycans [98]. In turn, the ECM has profound effects on tumor-cell invasion and immune-cell infiltration [99]. Since cancer has been compared with a “wound that would not heal” [100], it is tempting to hypothesize that ILCs may participate in melanoma development and progression by a reciprocal dialogue with other cells and molecules in the TME [101].

Several groups have correlated several ECM proteins with specific ILCs activities. In malignant-melanoma patients, an inverse association exists between the expression of hyaluronan and proteoglycan link protein-3 and NK-cell infiltration [102]. Heparan sulfate proteoglycans may act as NK-cell ligands but are not sufficient to induce NK cells’ cytotoxic activity against melanoma cells. Hypothetically, the overexpression of heparanase by melanoma cells or NK cells reduces the ability to recognize target cells [103,104], thus favoring their immune escape. Other ECM proteins, such as galectin1 and galectin3, can directly inhibit the activity of NK cells in some types of cancer, including melanoma [105,106,107]. While some galectins play a protumoral role, others such as galectin-9 exhibit antitumoral activity by modulating NK-cell activity [107].

There is ample evidence supporting the relevance of the interaction among ILCs and stromal cells in cancer development. In the tumor stroma, endothelial cells might regulate NK antitumor responses through NK group 2D (NKG2D) ligands. Thompson et al., suggest that the disruption of the NKG2D ligand, RAE-1ε, in B16 melanoma-associated endothelial cells impairs NK antitumoral responses in vivo [108]. Conversely, NK cells might modulate tumor vascularization through the production of proangiogenic molecules. Levi et al., demonstrated that in several cancers, including melanoma, the majority of NK-infiltrating cells are of the CD56^bright^ subset, which secretes the proangiogenic factor vascular endothelial growth factor (VEGF) contributes to the development on new vessels [109]. Additionally, NK cells are located near tumor-associated myofibroblasts in melanoma lesions, suggesting that their crosstalk may result in the induction of a proangiogenic phenotype of NK cells [110] while inhibiting the antitumor cytotoxicity of NK cells [110,111]. Cancer-associated fibroblasts induce the shedding of the NKG2D ligands major histocompatibility complex (MHC) class I chain-related protein A and B (MICA/B) expressed on the tumor-cell surface by secreting high levels of metalloproteases (MMPs). As a result, melanoma cells become more resistant to NK-cell-mediated lysis [112]. Moreover, cancer-associated fibroblasts decrease the expression of NK activating receptor through the secretion of prostaglandin E_2_ and indoleamine-2,3-dioxygenase (IDO), attenuating the cytotoxic activity of NK cells [110,113]. Thus, NK cells may participate in reactions known to occur in the tumor-associated stroma.

Other ILCs are also able to initiate complex crosstalk with stromal cells. For example, IL-33, constitutively produced by fibroblasts and endothelial cells in the skin, prompt Type 2 immune responses [114] by, among other actions, activating macrophages and ILC2 [115]. IL-33-activated ILC2s secrete IL-13 that induces the transdifferentiation into myofibroblasts. Other actions of ILC2-derived IL-13 are M2 polarization of tumor macrophages and the activation of MDSCs [101,116]. Both actions contribute to a microenvironment that supports tumor survival.

Type 3 ILCs might also influence the local immune response by interacting with the tumor stroma. Eisenring et al., showed that IL-12-activated NKp46^+^-expressing LTis increased leukocyte invasion by inducing the expression of intercellular adhesion molecule-1 (ICAM-1) and vascular cell adhesion molecule-1 (VCAM-1) in the tumor vasculature, Ref. [117]. Other Type 3 ILC3s may also act on stromal cells through IL-17, which induces VEGF, TGFβ, or IL-8 [118]. Moreover, ILC3-derived IL-17 might promote M2-macrophage differentiation and the recruitment of MDSCs [119,120].

### 4.2. ILCs in the Metastatic Progression of Cutaneous Melanoma

During melanoma progression, tumor cells acquire the ability to metastasize to distant organs, such as the liver and lungs. These new microenvironments regulate cell behavior within a complex stroma (Table 2).

#### 4.2.1. ILCs and Inflammation during Metastatic Progression in the Liver

The relationship between inflammation and tumorigenesis has led to the identification of tumor-related inflammation as one of the hallmarks of cancer [121]. In general, the liver presents a tolerogenic environment to avoid undesired immune reactions to oral antigens, which predisposes the liver to other types of pathogenic infections. As a result, the liver may develop fibrosis, and, in turn, hepatocellular carcinoma [122]. Moreover, hepatic fibrosis has been shown to predispose this organ to metastasis [123]. For example, invasion by metastatic cells may be facilitated by the inflammatory state that exists in the fibrotic liver, which promotes the adhesion of metastatic cells to the liver’s capillaries, inducing the further release of cytokines [39,124]. Inflammation also stimulates ILCs and recruits immunosuppressive immune populations to metastatic initiating sites [101,125].

Members of the IL-1 superfamily, such as IL-1β, IL-18, and IL-33, are involved in inflammatory diseases and cancer. In several cancers, including melanoma, the IL-1 superfamily modulates the TME to promote liver metastasis [126]. Less is known about the role of these cytokines in re-educating innate-lymphoid-immune-cell function in metastatic growth. IL-18 neutralization has shown opposing effects in controlling metastasis to the liver [127,128]. Indeed, some reports support IL-18-signaling-induced hepatic NK-cell maturation and, thus, a significant inhibition of tumor growth in the liver [39,128,129]. On the contrary, Salado et al. reported a protumoral role for IL-18 in the inflammation-dependent development of melanoma metastasis to the liver [130]. In the liver, IL-18 is produced by liver non-parenchymal cells and by myeloid dendritic cells (DCs) [131]. IL-18, in conjunction with IL-1β, contributes to the stimulation of IFNγ production by NK cells, which, along with granzyme, facilitates the generation of an inflammatory milieu [132]. Within this environment, NK cells promote the phagocytosis of colon carcinoma cells by liver-resident macrophage Kupffer cells (KCs) [133]. Even though the immune populations cooperate to mount an initial reaction against the tumor in the pre-metastatic niche, this protective response is soon reversed [117,134].

Besides their immunological role in the metastatic progression of melanoma to the liver, inflammatory cytokines have also been implicated in the adhesion of circulating murine melanoma cells to the hepatic microvasculature through ICAM-1 and VCAM-1 adhesion molecules [17]. Eisenrig et al. also showed that NKp46^+^ ILC3s were capable of upregulating both cell-adhesion molecules, facilitating the infiltration of immune cells with anti-tumor functions [117]. Further studies are needed to decipher the complex regulatory roles of ILCs in the metastasizing liver.

#### 4.2.2. ILCs and Inflammation during Metastatic Progression to Lungs

One of the main target organs for circulating melanoma cells is the lungs. Several types of ILCs have been related to melanoma progression to the lungs [42]. The tumoricidal activity of NK cells in the lungs is greatly enhanced by other inflammatory cells, such as DCs and macrophages (Table 2). Chiba et al. showed that the dectin-1 expressed on these innate immune cells contributes to the NK activation and effective killing of B16 melanoma cells during metastatic progression to the lungs [135]. Dectin-1 is also expressed by hepatic KCs [136], which indicates similar roles of lung and liver macrophages during metastatic progression. In murine lungs metastasized by B16F10 melanoma, fully active and highly cytotoxic NK cells produce large amounts of IFNγ [137]. Such an increase in IFNγ is associated with a more effective immune response and an increased in fibronectin, an ECM protein involved in the development of a non-permissive tumor environment [15]. Moreover, a defect in NK-cell differentiation supports the anti-metastatic role of NK cells in melanoma metastasis to the lung [138]. Leong et al. have reported an increase in the lung metastasis of B16-F10 melanoma cells in mice carrying an NK-specific phosphatase and tensin homolog (PTEN) deletion [139]. Despite the role of PTEN in limiting the cytotoxic activity of NK cells, the prometastatic effect was attributed to a direct effect of PTEN deletion on NK-cell trafficking [140].

In the lungs, ILC2s increase in number during inflammatory diseases and mediate the production of IL-1β, IL-18, IL-33, IL-5 and IL-13 [141]. In this scenario, the IL-33-mediated activation of ILC2 cells might control the development of metastasis by promoting tumor immune-surveillance through eosinophil infiltration due to ILC2-derived IL-5 and IL-13 [142]. Moreover, ILC2 might give rise to IFNγ-producing ILC1s [42] and the potential for transdifferentiation into other ILC types, introducing increasing levels of complexity in the study of ILC2s’ role during tumor progression.

The IL-1 superfamily members regulate the production of IL-17 by ILC3s [143]. In the lung, galectin-3 facilitates melanoma metastatic colonies by affecting tumor-cell adhesion and the innate immune response against melanoma by increasing serum IL-17 levels [106]. In melanoma metastasis to the lung, IFNγ, in combination with IL-17, models the TME in such a way that an effective immune response to the tumor can operate [106]. However, it is unknown whether IL-17 derives from naturally occurring ILC3 or from ILC1-derived NCR(+)ILC3s favoring tumorigenesis [144].

#### 4.2.3. ILCs and the Stromal Compartment in the Metastatic Microenvironment

During metastatic progression, and similarly to primary-tumor development, ILCs interact not only with tumor cells but also with fibroblasts and endothelial cells, key stromal components with central roles in the generation and remodeling of the TME. Little is currently known regarding the specific role of different ILCs in the modulation of the stromal compartment that ultimately leads to the spread of melanoma to the liver and lungs. What is known and clear is that the adhesion of tumor cells to the endothelium of organs is essential for metastasis development [145]. ILCs enhance endothelial cells’ expression of VCAM-1 and ICAM-1. It is interesting to note that VCAM-1 [146] and ICAM-1 [147] play critical roles in the colonization of the liver and brain melanoma metastasis, respectively [146,147,148,149]. Moreover, ILC3s promote the adhesion and recruitment of leukocytes to the sites of invasion [91]. ILC2 trafficking is regulated by β2 integrin expression in the context of the inflammatory milieu [150], raising the possibility that high ICAM-1 expression on the lung and liver endothelium favors the recruitment of ILCs during melanoma metastasis.

The metastatic TME in the liver contains soluble mediators that recruit fibroblasts, pericytes, and stem cells to the site of tumor colonization and induce their transdifferentiation into myofibroblast-like cells. Such cells are characterized by their expression of α-smooth muscle actin stress fibers, a high proliferation rate, and a large production of fibrillary collagen [151]. Unresolved fibrosis that persists during metastatic growth sustains inflammation, which promotes the recruitment of ILCs to the developing metastatic foci. Interestingly, ILCs modulate fibroblast function by driving the secretion of excessive collagen and matrix proteins in the lung and the liver [152,153].

Amphiregulin is considered a potential prognostic marker for liver metastasis in colorectal cancer progression and a mediator of liver-myofibroblast activation [154]. This relationship has not been addressed in the context of melanoma progression. However, given the implication of ILCs in fibrogenesis, inflammation, and liver disease, it is tempting to hypothesize that ILC2s play a prometastatic role in the liver in melanoma, as they do in colorectal cancer [155], by secreting factors that promote the development of a desmoplastic stroma [156].

Opposing roles of ILC3s are observed between the early metastatic stages and once their interaction with cancer-associated fibroblasts has taken place. Early in tumor development, the increasing numbers of ILC3s create an environment more protective against melanoma metastasis. Carrega et al., (2015) showed that NCR(+)ILC3s are activated by tumor-associated fibroblasts through the NKp44 activating receptor [157]. The activation of lung fibroblasts results in the production of large amounts of insulin-like growth factor 1 (IGF-1) [158], which favors ILC3 recruitment and activation. However, the IGF-1 level has also been correlated with melanoma lung-metastasis development and the appearance of acquired chemoresistance [159].

## 5. ILCs and EVs in the TME

An updated view of the TME includes not only “free” molecules or direct communication mediated by cell-to-cell contact but also vesicle-driven systems [160]. The generic term “extracellular vesicles” (or “EVs”) encompasses all types of membrane-surrounded vesicles released from cells into the extracellular medium. The capacity of cells to secrete vesicles has been known since the late 1960s [161], although it was not until the late 1990s that EVs were included among the mechanisms driving cell-to-cell communication [162,163]. These pioneering studies described the involvement of secreted vesicles in the communication among immune cells and in the antitumor activity exerted by DCs, placing the immune system in the foundations of the EV research field. Moreover, increasing evidence shows a central role for EVs in the protumoral and prometastatic remodeling of the TME by sustaining cell proliferation and reprogramming stromal cells, in addition to the abovementioned role in the immune response [160].

Size and cellular origin-based classifications distinguish three main classes of EVs: (1) apoptotic bodies, 80–5000 nm-sized vesicles formed during the programmed death of a cell (sometimes included in the second group described here); (2) microvesicles, ectosomes, microparticles, or oncosomes (when related to the oncology field), which are described as 50–1000 nm-sized vesicles generated by the outward budding of the plasma membrane; and (3) exosomes, vesicles originated in the endosomal system with an approximate diameter of 50–150 nm and liberated into the extracellular medium by the fusion of multivesicular endosomes (MVE) with the plasma membrane. EVs may carry different cargoes, including proteins, lipids, and nucleic acids, and represent a stable strategy for transferring otherwise-unstable molecules such as mRNAs [160,164,165]. Nevertheless, EVs from different modes of biogenesis often display similar appearances, overlapping size ranges, and comparable compositions; therefore, unless an exhaustive isolation protocol is applied, it is undoubtedly challenging to clarify the true nature of the EVs with which we are working. We, therefore, adopt this general term to discuss the current knowledge regarding secreted vesicles in the TME and primarily related to the ILCs.

The EV field has garnered enormous interest over the past few years due to its involvement in the short—and long-distance organ communication system as well as a putative drug-delivery system. It is especially relevant for highly spreading malignancies, such as melanoma, where few therapeutic options exist for advanced stages. In this context, EVs have shown to play an essential role in modulating both TME and distant-organ behavior [166,167,168,169]. Considerable effort has been devoted to elucidating the immunomodulatory capacity of EVs and, especially, the role of tumor-derived EVs in the formation of both a limiting and a favorable environment for cancer progression [160,170]. The first studies in this field pointed towards an immunoreactive role for tumor-derived EVs. As shown by Wolfers et al., EVs secreted by tumor cells were able to induce antitumoral activity against mammary adenocarcinoma, colon adenocarcinoma, and mastocytoma tumor models [171]. In the same study, they observed the transference of melanoma antigens (e.g., Mart-1) present on EVs to DCs, which was suggested as the mechanism underlying the observed activation of cytotoxic T lymphocytes (CTL). In agreement with the previous study, Dai et al. observed a specific CTL response driven by tumor-antigen-containing EVs [172]. The immune- suppressive effects of tumor EVs have also been described. T-cell apoptosis driven by Fas ligand (FasL), TRAIL, or galectin 9-containing EVs; the inhibition of DC/macrophage maturation by TGFβ-dependent mechanisms; and monocyte differentiation into MDSCs are among the described strategies for silencing the antitumor activity of the host [170,173].

### 5.1. Effect of Tumor-Derived EVs on ILCs

Focusing on ILCs, the current knowledge is circumscribed to NK cells and to a period approximately spanning the past fifteen years. The limited information thus available prompted us to review EVs behind the frontiers of melanoma.

In 2005, Gastpar et al., reported that heat shock protein-70 (Hsp70)-positive tumor EVs activate NKs. As explained in this work, EVs, which contain Hsp70, were able to activate not only the migratory but also the cytolytic activity of NK cells against Hsp70-membrane positive tumors (Figure 6A) [174]. Hsp70 is broadly present in tumor-cell-derived EVs, including those from melanoma cell lines such as B16 [158]. The overexpression of Hsp70 on melanoma cells also induces NK-cell activation and antitumor activity against cells expressing NK-receptor ligands [173]. Nevertheless, caution must be taken, as exosomal Hsp70 has also been linked to MDSC activation, which may lead to an immunosuppressive TME [175,176,177]. In 2006, Liu et al., observed that the pretreatment of mice with EVs originating from mammary tumors accelerated subsequent tumor growth. According to this study, the protumoral activity was partially explained by diminished proliferation of the tumor cells and the cytotoxic capacity of NK cells [178].

The regulation of NK-cell receptors could be one of the mechanisms behind the altered NK activity. Interestingly, the presence of activating surface-receptor ligands on EVs has been proposed to act as both an inducer and repressor of NK-cell activity (Figure 6B,C) [179,180]. HLA-B-associated transcript 3 (BAT3), a ligand for the NKp30 receptor [180], can be secreted in exosome-like vesicles from tumor cells and induce IFNγ and TNFα release by NKs. Notably, the incubation of NKs with purified BAT3 exerts an opposite effect, suggesting that membrane-bound, and not soluble, BAT3 is required to elicit NK-mediated cytotoxicity. The nature of BAT3-containing vesicles was promptly confirmed by Simhadri et al. [181].

The presence of NKG2D ligands on EVs seems to represent a double-edged sword; while they promote NK activation and proliferation in some experimental contexts [182], NKG2D ligands may also suppress NK-cell cytotoxicity by behaving as a decoy [178,180,181]. The observed differences may be linked to the origin of the ligand-containing EVs, as DC-derived EVs lead to immunoreactivity [182], while exposing NKs to tumor-originated EVs leads to immunosuppression [180,183]. Interestingly, EVs obtained from certain melanoma cell cultures also contain NKG2D-ligand MICA, and these EVs can diminish the surface expression of the receptor on NKs [184]. The same immunosuppressive effect of melanoma-derived EVs was also recently described by a study focused on the characterization of melanoma-patient-derived EVs [185].

TGFβ represents a second primary mechanism for NK modulation. As beautifully reviewed by Batlle and Massagué, TGFβ is an essential regulator of immune homeostasis and also plays a relevant role in tumor immune evasion [186]. The TGFβ-mediated inhibition of NK cells includes the downregulation of surface NKG2D and NKp30 receptors, decreased receptor adapter DAP12 levels (Figure 6D), and the modification of metabolic mammalian target of rapamycin (mTOR) signaling. TGFβ has also been shown to be present on tumor-derived EVs and to play a crucial role in NK-cell activity, at least partly by downmodulating surface receptors [187,188]. In a more biological setting, EVs purified from the sera of patients who have acute myeloid leukemia (AML) and pancreatic cancer contain TGFβ [187,188]. Furthermore, and despite a relatively low number of serum EV samples, both studies were able to detect higher amounts of TGFβ in AML patients and describe the role of this growth factor in the inhibition of NK-cell activity [189].

Interestingly, melanoma patients exhibit higher blood TGFβ levels [190,191], although the specific presence of TGFβ in blood-derived EVs remains elusive. TGFβ has recently been detected in melanoma-cell-derived EVs [190], and the results support a role for EV-derived TGFβ in the downregulation of DC-receptor molecules (e.g., CD40 and CD86). Despite the lack of specific data regarding the effects of melanoma-derived TGFβ on NK-cell activity, it seems plausible to speculate about its role in immunosuppression also by NK-cell inhibition.

### 5.2. Tumor Modulation by ILC-Derived EVs

As previously mentioned, the immune system must eliminate “dangerous foreigners”, including cells undergoing malignant transformation. Any failure in this mission facilitates cancer development. Innate immune cells, including NK cells, are part of the first line of defense that elicits non-specific but fast antitumoral activity. NK-mediated cancer-cell destruction relies on a delicate balance among activating and inhibiting cell-surface receptors. An outcome favoring activating receptors (e.g., NKG2D, NKp30, NKp46, and DNAM-1) leads to the release of cytotoxic effectors (e.g., perforin, granzymes A and B, and FasL) and the death of the target cell.

Interestingly, the EVs secreted by NK cells contain functional immunomodulatory surface molecules as well as lytic proteins, providing a plausible explanation for their observed antitumoral activity [192,193,194,195,196]. In detail, Lugini et al. identified NK markers (mainly CD56 and NKG2D) in EVs isolated from healthy-donor blood samples as well as perforin and FasL, proteins inducing the cell death of the NK-target cells [193]. Moreover, in the same study, they found that activated NK-derived EVs were able to exert cytolytic effects against several hematological cell lines, although little effect was observed against breast and melanoma cell lines. Jong et al. also identified CD56 as a component of the EVs obtained from activated NK cultures derived from human blood [194]. They also detected several lytic proteins (perforin, granulysin, granzyme (A), and granzyme (B), which could explain the cytotoxic activity of EVs against leukemia, neuroblastoma, and breast-cancer cell lines. Shoae-Hassani et al. extended the catalog of NK receptors found in NK-derived EVs, while perforin and FasL were identified among the contents of EVs purified from NK-92MI cell cultures [195].

Limited and contradictory literature exists regarding the specific role of ILC-derived EVs in melanoma. While little antitumor activity was detected for EVs from blood-derived NK cells [180], in vitro and in vivo cytotoxicity was detected by Zhu et al. against B16F10 melanomas treated with EVs obtained from NK-92MI cells [196]. These conflicting results may indicate that diverse results can be obtained when using different sources of NK-derived EVs or melanoma cell lines and conditions. In this particular scenario, it is important to consider the limitations to the expansion and activation of blood-derived NK cultures, compared to the use of an IL-12-overexpressing transformed NK-cell line (NK-92MI) that exhibits continuous expansion in culture. Moreover, the amount of EVs applied is also a variable to consider, although higher amounts are claimed to be employed in non-effective assays than those showing antitumor activity. Jong et al. proposed a method to enlarge the scale of EVs obtained from blood-derived activated human NK cells and analyzed the cytotoxicity against several non-melanocytic tumors [194]. The results suggest a path that could be explored in melanoma models. In addition, NK-cell “education” may essential for the selective EV-mediated induction of tumor-cell death. In this regard, Shoae-Hassani et al. reported increased anti-tumor activity for blood-derived cytokine-activated NK cells when they were incubated with the target neuroblastoma cells before EV isolation from the NK cultures [195].

## 6. Conclusions

The discovery of ILCs and their roles in immunity, inflammation, tissue repair, and maintenance has significantly raised interest in these populations and their role in tumor progression. In cancer, the currently available data report contradictory and conflicting results that reflect the complexity of the field (Table 2). Such complexity may be partially driven by the reported capacity of ILC subsets to transdifferentiate in response to the TME and homing tissue. For example, increased numbers of ILC1s might result from the transdifferentiation of ILC3s into helper ILC1s or from the conversion of NKs. Nevertheless, mentioned differences may also result from the lack of broadly stablished consensus on ILC population-specific markers. In fact, discrimination of ILCs from other immune populations require a complex set of markers, nor say the identification of specific ILC types. Time and effort are required to settle down knowledge specially on complex fields such as lineage tracing.

Determination of most meaningful ILC populations for primary or metastatic melanoma development may enormously benefit from advanced tissue analysis based on multiplexing tissue imaging while involvement of ILCs on treatment response could be approached by serum or plasma analysis. These could represent solid foundations for further analysis involving specific ILCs and their interaction with melanoma cells either by ILC isolation and culture with syngeneic melanoma cells or by more complex animal models that should be developed. Besides the direct action of ILCs on the tumor cell, mounting evidence suggests that ILCs can influence the activity of other stromal cells through the promotion of immune editing, angiogenesis, and ECM remodeling. Correlation among these processes and specific ILC populations could be deciphered by in situ tissue analysis while proper functional analysis would require models able to recreate TME.

The level of complexity in the characterization of ILCs and their role in tumors increases when considering the critical role of an additional component of the TME: the extracellular vesicles. The available information on ILC-derived EVs remains scarce and somewhat contradictory. While little antitumoral activity has been detected for EVs from blood-derived NKs, the cytotoxic potential of EVs obtained from NK-92MI cells for B16F10 melanoma is high both in vivo and in vitro. Given the influence EVs exert on a large plethora of pathways and mechanisms, it is necessary to increase our knowledge on the effects of EVs from ILCs to better focus the development of effective therapies. Nevertheless, the development of this field encounters on the cell expansion a limitation; isolation of EVs by classical methodologies such as differential ultracentrifugation of chromatography-based methods requires large number of cells, especially for further functional analysis. Current efforts are focused on protocols to allow NK cell-expansion as an alternative to the use of transformed cell lines such as NK-92 and NK-92MI. Similar efforts will be required in order to analyze the effect of other ILCs in tumor, and specifically melanoma, development.

Therefore, the identification of markers and tools to allow the modulation of individual ILC subsets in mice and humans, the development of standardized protocols, the deep characterization of melanoma-related ILCs and the identification of their key contribution to cancer development will be essential prior addressing the therapeutic modulation of ILCs.

## Figures and Tables

**Figure 1 cancers-12-03177-f001:**
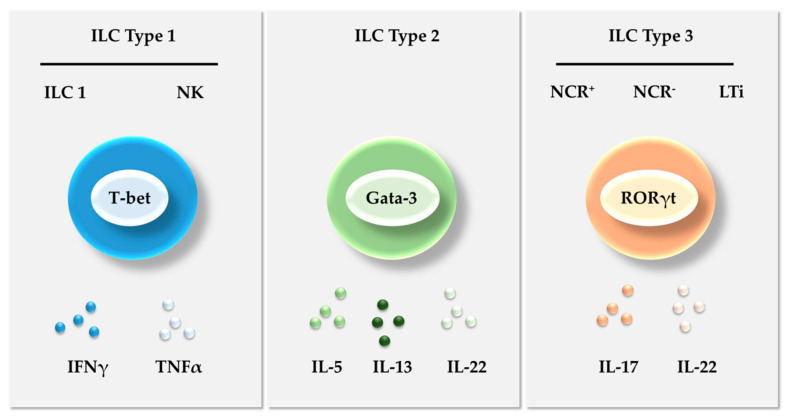
Innate lymphoid cell classification. The ILCs are classified into three main groups—Type 1, Type 2, and Type 3 ILCs—according to the differential expression of transcription factors and specific cytokine repertoires that they produce. NCR—Natural cytotoxicity receptors; IFNγ—Interferon gamma; TNFα—Tumor necrosis factor alpha.

**Figure 2 cancers-12-03177-f002:**
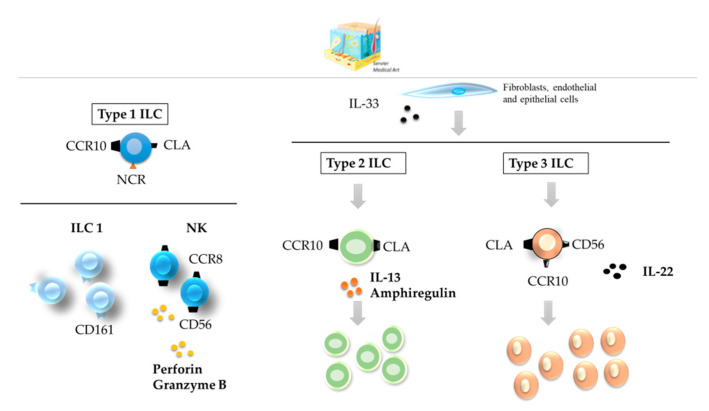
ILCs in the healthy skin. All three types of ILCs can be found in healthy skin; CD56^+^ CD16^−^ NK cells, ILC2s, and NCR(−) ILC3s are the most abundant. All types of skin ILCs express the surface receptors CCR10 and CLA. ILC—innate lymphoid cell; NCR—NK cell receptor; CCR—C-C motif chemokine receptor; CLA—cutaneous lymphocyte-associated antigen.

**Figure 3 cancers-12-03177-f003:**
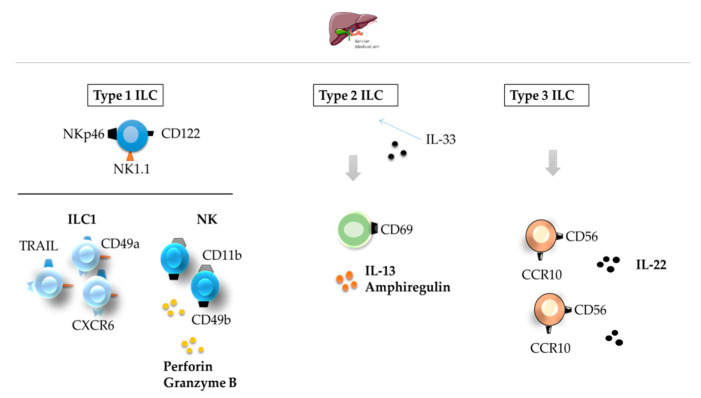
ILCs in the healthy liver. All three types of ILCs can be found in the healthy liver. Type 1 ILCs are the most abundant ILC type found in the healthy liver. Type 2 and 3 ILCs represent only about 5% of all ILCs. ILC—innate lymphoid cell; CCR—C-C motif chemokine receptor [51,52].

**Figure 4 cancers-12-03177-f004:**
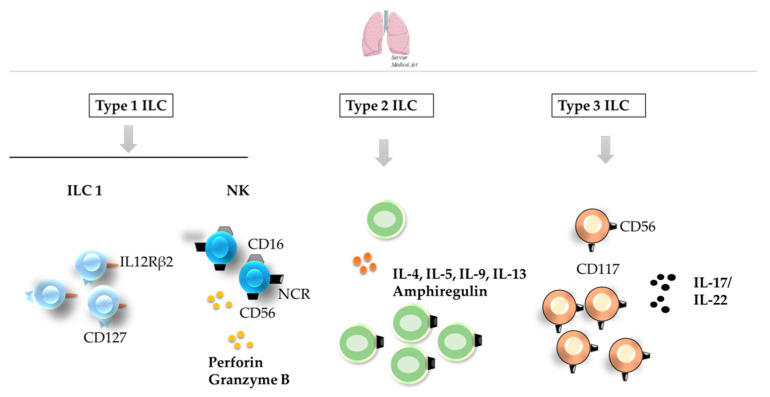
ILCs in the healthy lung. Although ILC1 and NK cells (**left**) can also be found in healthy lung, ILC2s (**middle**) and NCR(−) ILC3s (**right**) are the predominant subsets in healthy human lungs. Some of the receptors employed for the identification of different ILC subsets are shown in the image. ILC—innate lymphoid cell; NCR—NK cell receptor.

**Figure 5 cancers-12-03177-f005:**
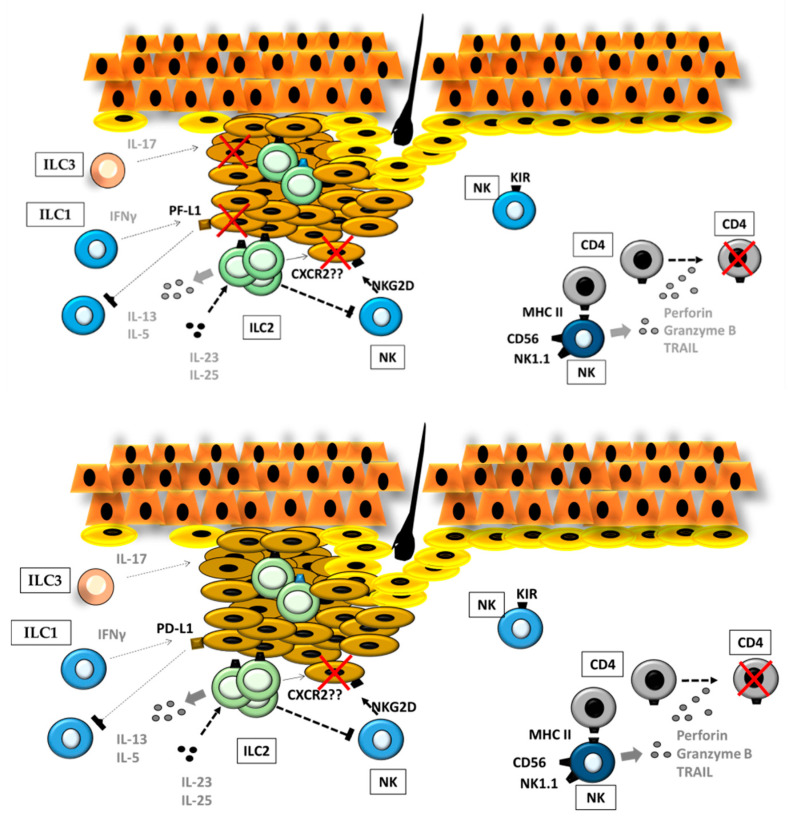
ILCs in melanoma. During melanoma progression, different ILC subsets maintain active crosstalk, resulting in either antitumoral or protumoral responses, not only by direct action on tumor cells but also by acting on other ILCs (left). Moreover, ILCs modulate CD4^+^ T-cell activity and, thus, influence the adaptive immune response through MHC class II. MHC—Major histocompatibility complex.

**Figure 6 cancers-12-03177-f006:**
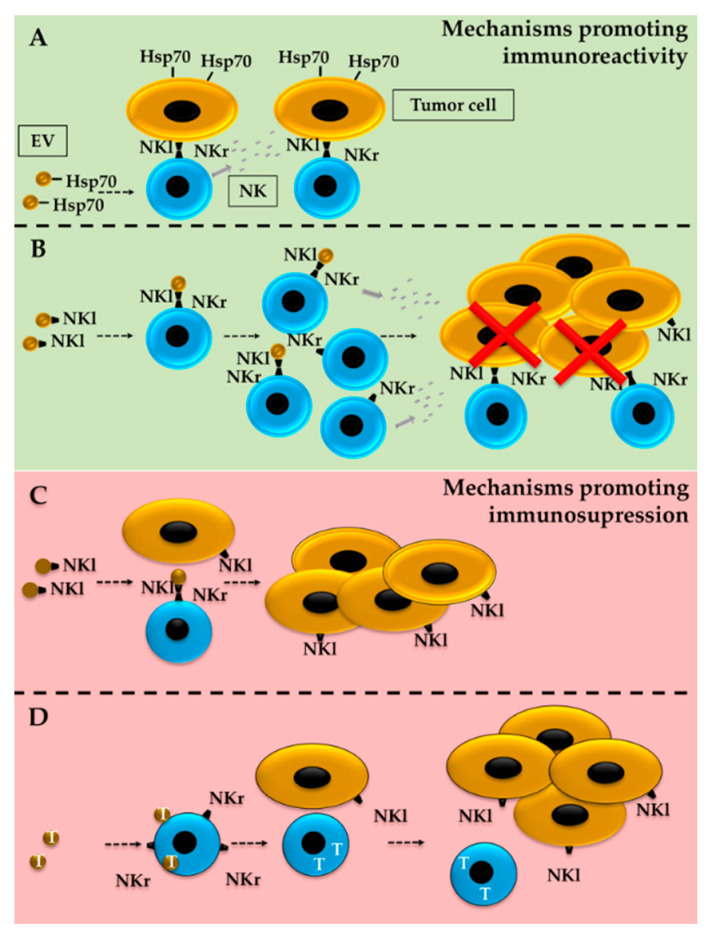
Regulation of ILC activity by tumor-derived EVs. Exposure of NK cells to tumor-derived EVs may lead to increased anti-tumor activity but also decreased immunoreactivity. The presence of Hsp70 (**A**) or ligands that activate the NK receptor (NKl) (**B**) have been proposed as mechanisms driving NK-cell activation, proliferation, and tumor-cell death. Nevertheless, the presence of NKls on tumor-secreted EVs may also lead to the opposite effect (**C**) if behaving as a decoy for the activation of NK receptors (NKr). EVs carrying TFGβ (T) (**D**) decrease NKr content on NK cells, which blocks NK-cell activation as well as the recognition of NKl-presenting tumor cells.

**Table 1 cancers-12-03177-t001:** Relative abundance of ILCs in skin, liver and lung.

		Skin		Liver		Lung	
		Healthy	Melanoma	Healthy	Melanoma	Healthy	Melanoma
Type 1 ILC	ILC1	+	++	++	+	++	NA *
	NK	++	++++	+++	++++	++	++++
Type 2 ILC		++++	+++	++	++	+++	++++
Type 3 ILC	NCR(+)	+	+++	+	++	++	+++
	NCR(−)	+++	+	+	++	+++	NA *

* Not sufficiently addressed. ILC—Innate lymphoid cells; NCR—Natural cytotoxicity receptor; NK—Natural killer.

**Table 2 cancers-12-03177-t002:** Functions of ILC types within a normal and tumor microenvironment during melanoma progression.

		Skin		Liver		Lung	
		Healthy	Melanoma	Healthy	Melanoma	Healthy	Melanoma
Type 1 ILC	ILC1	Immune cell recruitment.	Antitumoral (antigen recognition), protumoral (PD-L1 expression).	Normal hepatic function, local immune tolerance.	Inflammatory tumor microenvironment, tumor growth.	Homeostasis, host protection, immune surveillance.	Antitumoral (IFNγ-production)
	NK	Cytotoxicity.	Antitumoral (cytotoxic), protumoral (T cell exhaustion, ILC2 modulation).	Local immune response.	Antitumoral (cytotoxicity).	Cytotoxicity.	Antitumoral (cytotoxicity).
Type 2 ILC		Wound healing.	Antitumoral (IL-5), protumoral (NK cell-impairment, fibroblast transdifferentiation, increase NK PD-1 expression).	Response to chronic stress, local immune-suppression, counteract inflammatory injury.	Antitumoral (NA *), protumoral (favour a protumoral desmoplastic reaction)	Host protection, immune surveillance.	Antitumoral (tumor immunesurveillance, induction of IFNγ producing ILC1), protumoral (NK cell imapirment).
Type 3 ILC	NCR(+)	Immune cell recruitment.	Proangiogenic, immunesuppressive cell recruitment, IL-6 and Stat-3 activation	Response to chronic stress, counteract inflammatory injury.	Upregulation of adhesion molecules	Regulation of inflammation.	Antitumor immune response
	NCR(−)	Wound healing.	Proangiogenic, immunesuppressive cell recruitment, IL-6 and Stat-3 activation	Response to chronic stress, counteract inflammatory injury.	NA *	Regulation of inflammation.	NA *

* NA: Not addressed.

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
