# Peer review of "Innate Lymphoid Cells in the Malignant Melanoma Microenvironment"

_cancers, 2020, doi:10.3390/cancers12113177_

Round 1
Reviewer 1 Report
This is a very nice review of ILCs in melanoma. The authors addressed all issues, no further changes needed.
Author Response
We thank the reviewer for the time dedicated to reviewing the manuscript.
Reviewer 2 Report
In this review, the authors summarized the functional roles and inflammatory mechanisms of ILCs in melanoma. Based on stages of melanoma, the authors provided an informative summary on different ILCs, in particular focusing on NK cells in inflammatory responses, EVs as well as interaction towards stromal cells in primary and metastatic tissue microenvironment. Overall, the review is informative and attractive to potential researchers.
Major points:
- An additional description of limitations and future perspectives in the field of ILCs for melanoma needs to be further expanded in the conclusion/discussion section.
- As a type of cancer that showed a relatively higher responding rate to immunotherapy, especially immune checkpoint blockade therapy, it is interesting to add some points of ILCs and tissue microenvironment related to immunotherapy in melanoma.
- Figures 2-4 contained similar information of ILCs in skin, liver and lung. Could the authors provide some comparisons, either as a joint figure or a table to discuss the similarity and differences of ILCs in different tissues that related to primary or metastatic melanoma?
Minor point:
Figure 3, should be ‘healthy liver’ instead of ‘ healthy skin’.
Author Response
Point 1: An additional description of limitations and future perspectives in the field of ILCs for melanoma needs to be further expanded in the conclusion/discussion section.

Response 1: As requested by the Reviewer, we had added a little piece of information, reading as follows:
- Lines 600-615: “Nevertheless, mentioned differences may also result from the lack of broadly stablished consensus on ILC population-specific markers. In fact, discrimination of ILCs from other immune populations require a complex set of markers, nor say the identification of specific ILC types. Time and effort are required to settle down knowledge especially on complex fields such as lineage tracing.
Determination of most meaningful ILC populations for primary or metastatic melanoma development may enormously benefit from advanced tissue analysis based on multiplexing tissue imaging while involvement of ILCs on treatment response could be approached by serum or plasma analysis. These could represent solid foundations for further analysis involving specific ILCs and their interaction with melanoma cells either by ILC isolation and culture with syngeneic melanoma cells or by more complex animal models that should be developed. Besides the direct action of ILCs on the tumor cell, mounting evidence suggests that ILCs can influence the activity of other stromal cells through the promotion of immune editing, angiogenesis, and ECM remodeling. Correlation among these processes and specific ILC populations could be deciphered by in situ tissue analysis while proper functional analysis would require models able to recreate TME.”
- Lines 623-632: Nevertheless, the development of this field encounter on the cell expansion a limitation; isolation of EVs by classical methodologies such as differential ultracentrifugation of chromatography-based methods requires large number of cells, especially for further functional analysis. Current efforts are focused on protocols to allow NK cell-expansion as an alternative to the use of transformed cell lines such as NK-92 and NK-92MI. Similar efforts will be required in order to analyze the effect of other ILCs in tumor, and specifically melanoma, development.
Therefore, the identification of markers and tools to allow the modulation of individual ILC subsets in mice and humans, in addition to the development of standardized protocols, the deep characterization of melanoma-related ILCs and the identification of their key contribution to cancer developmenti s will be essential for prior addressing the therapeutic modulation of ILCs.
Point 2: As a type of cancer that showed a relatively higher responding rate to immunotherapy, especially immune checkpoint blockade therapy, it is interesting to add some points of ILCs and tissue microenvironment related to immunotherapy in melanoma..
Response 2: As suggested by the reviewer, we had added some pieces of information regarding immunotherapy in lines 41-43, 211-213 and 238-240.
Point 3: Figures 2-4 contained similar information of ILCs in skin, liver and lung. Could the authors provide some comparisons, either as a joint figure or a table to discuss the similarity and differences of ILCs in different tissues that related to primary or metastatic melanoma?
Response 3: As requested by the reviewer we have added 2 tables shown relative frequencies and comparative function in healthy and tumoral conditions.
Point 4: Figure 3, should be ‘healthy liver’ instead of ‘healthy skin’.
Response 4: We thank the reviewer for pointing the mistake in the figure legend. The error has been corrected.
Reviewer 3 Report
Authors review several (if not all) aspects of innate lymphoid cells in melanoma tumor micro-
Authors review several (if not all) aspects of innate lymphoid cells in melanoma tumor micro-environment (and more). In general, it is a well written and complete overview. Content-wise I have no comments or suggestions. It is very complete.
Major remarks
However, it is a very lengthy manuscript. The “brief” description to innate lymphoid cells ( start line 50) as well as the “brief” preliminary description (line 111) actually only end at line 192. Furthermore, the sentences used in the text are often very long and similar topics are highlighted on different sections (e.g. “as previously mentioned” itself is already three times present in the text) . Authors may consider reducing the size of the manuscript substantially and splitting long sentences in two (or three….).
Although this is a review, it misses a clear topic /message (and subsequently a clear conclusion) The titel, abstract and introduction give the impression its about the role of ILCs in tumor micro-environment. The lengthy second part (till line 192) reduce the focus of the manuscript.
Minor remarks
Graphics and tekst in the figures 1, 2, 3 and 4 are of insufficient quality (they are descriptive, but their image resolutaion/quality is not sufficient)
Reference 1 is not complete.
A lot of references (over 200) , it is not always necessary to site two manuscripts.
The readability of the manuscript would benefit from a table in which the different types of ILCs and their locations and roles are described for primary / metastatic melanoma,
Author Response
Response to Reviewer 3 Comments
Point 1: However, it is a very lengthy manuscript. The “brief” description to innate lymphoid cells (start line 50) as well as the “brief” preliminary description (line 111) actually only end at line 192.
Furthermore, the sentences used in the text are often very long and similar topics are highlighted on different sections (e.g. “as previously mentioned” itself is already three times present in the text). Authors may consider reducing the size of the manuscript substantially and splitting long sentences in two (or three….).
Response 1: As suggested by the reviewer, we have reduced the above section by 1/3 approximately. In addition, in an attempt to follow the suggestions to reduce the size of manuscript, “as previously mentioned-2 and its following sentences have been either erased or rewritten in more integrated way. Also, we have split many long sentences all through the text. As a result, some words have been either added or eliminated to convey with the new format of the paragraph.
Finally, we have reduced the size of the manuscript substantially from its initial 1226 lines to its final 1150, taking in account the additional information included as suggested by reviewer 2.
Point 2: Although this is a review, it misses a clear topic/message (and subsequently a clear conclusion) The title, abstract and introduction give the impression it’s about the role of ILCs in tumor micro-environment. The lengthy second part (till line 192) reduce the focus of the manuscript.
Response 2: We have reduced the second part to focus the topic of the manuscript. on the interplay between ILCs and the components of the tumor environment during the development of primary melanoma and metastatic progression. Being a review on a subject with such an ample field for research, and so many aspect still remain unknown, t´s hard to give a specific conclusion. Despite this, we have added some information in “Conclusion” as requested also by Reviewer 2.
Point 3: A lot of references (over 200), it is not always necessary to cite two manuscripts.
Response 3: We have removed 18 references from the previous version.
Point 4: Graphics and tekst in the figures 1, 2, 3 and 4 are of insufficient quality (they are descriptive, but their image resolutaion/quality is not sufficient)
Response 2: We thank the reviewer to take our attention about the quality of the graphics. We will increase the resolution to try improving their quality.
Point 5: Reference 1 is not complete.
Response 5: We thank the reviewer to take our attention about ref.1. We have corrected it.
Point 6: The readability of the manuscript would benefit from a table, in which the different types of ILCs and their locations and roles are described for primary / metastatic melanoma,
Response 6: As requested by the reviewer we have added 2 tables in an attempt of showing the relative frequency and roles of ILCs during melanoma progression.